# Research and Development on Cold-Sprayed MAX Phase Coatings

Weiwei Zhang [ID], Shibo Li *, Xuejin Zhang and Xu Chen

Center of Materials Science and Engineering, School of Mechanical and Electronic Control Engineering,
Beijing Jiaotong University, Beijing 100044, China; 22110428@bjtu.edu.cn (W.Z.); 22121382@bjtu.edu.cn (X.Z.);
22126130@bjtu.edu.cn (X.C.)
* Correspondence: shbli1@bjtu.edu.cn; Tel.: +86-10-51685554

**Abstract:** Cold spraying is an attractive solid-state processing technique in which micron-sized solid particles are accelerated towards a substrate at high velocities and relatively low temperatures to produce a coating through deformation and bonding mechanisms. Metal, ceramic, and polymer powders can be deposited to form functional coatings via cold spraying. MAX phase coatings deposited via cold spraying exhibit several advantages over thermal spraying, avoiding tensile residual stresses, oxidation, undesirable chemical reactions and phase decomposition. This paper presents a review of recent progress on the cold-sprayed MAX phase coatings. Factors influencing the formation of coatings are summarized and discussions on the corresponding bonding mechanisms are provided. Current limitations and future investigations in cold-sprayed MAX coatings are also listed to facilitate the industrial application of MAX phase coatings.

**Keywords:** cold spray; MAX phases; coatings; microstructure; properties; applications

## 1. Introduction

MAX phases (M is an early transition metal, A is an A-group element, and X is carbon, nitrogen or boron) are ternary-layered carbides, nitrides, and borides [1–4]. MAX phases have a hexagonal crystal structure consisting of MX slabs interleaved with a single "A" layer. The M-X bonds in the MX slabs are strong covalent bonds, but the bonds between the MX and A layers are relatively weak. This special layered-structure endows MAX materials with a combination of the attractive properties of ceramics and metals, such as self-lubrication, self-healing, machinability, high thermal and electrical conductivities, superior resistance to corrosion and oxidization, and nonsusceptibility to thermal shock [1–5]. To date, MAX materials have been successfully developed into products such as heating elements, gas burner nozzles, electrical contacts, pantographs, and thread bolts (Figure 1).

In addition to the above applications, MAX phases can be used as protective coatings against oxidation, corrosion, and friction. So far, MAX phase coatings, with thicknesses ranging from tens to hundreds of micrometers, have been mainly prepared via the feasible or cost-effective techniques of thermal spraying and cold spraying.

Thermal spraying techniques use thermal energy to melt and soften particles, which are then sprayed on a substrate using process gases to form a coating. According to the different heat sources used, thermal spraying techniques are divided into plasma spraying techniques, electric arc spraying techniques, and high-velocity oxy-fuel (HVOF) spraying. Among the above techniques, the HVOF technique is generally used to prepare MAX phase coatings due to its employment of a median heat energy (2100–3000 °C) [6–11]. For example, Frodelius et al. [6] prepared a $Ti_2AlC$ MAX coating with a thickness greater than 100 μm on a stainless-steel surface using the HVOF technique. The results showed that the $Ti_2AlC$ coating adhered to the substrate well. Chen et al. [7] fabricated a $Cr_2AlC$ MAX coating with a thickness greater than 200 μm using supersonic flame spraying. Thermally-sprayed MAX phase coatings have a dense microstructure and offer good adhesion to a substrate.

However, the HVOF technique requires the use of high temperatures (2100–3000 °C), which causes the oxidation and even decomposition of MAX phases. The oxidation or decomposition of MAX phases results in impurities with a high content in the resultant coatings, thereby deteriorating the functional performance of MAX coatings compared to bulk MAX phases. In addition, tensile stresses are inevitably generated in thermally-sprayed MAX coatings, inducing the crack formation and even the peeling of coatings. Therefore, achieving large-scale MAX coatings and retaining the original composition and properties of MAX phases through thermal spraying techniques remains challenging.

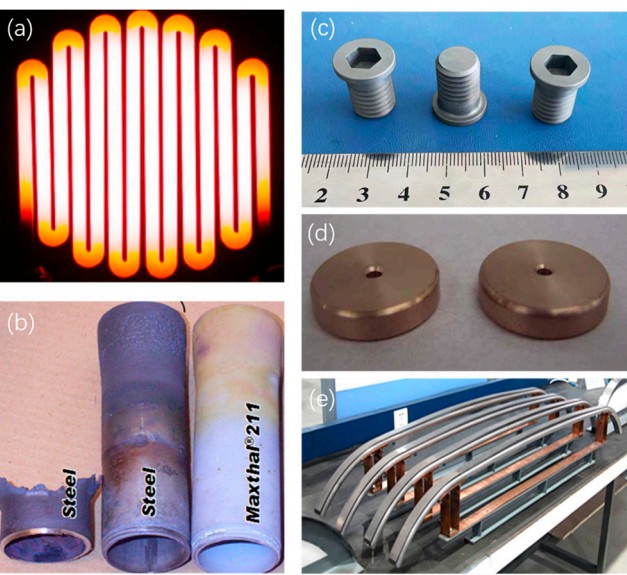

**Figure 1.** MAX products. (**a**) Maxthal 211 heater glowing at 1723 K (courtesy 3-ONE-2, LLC), (**b**) gas burner nozzles made of Maxthal 211 and 353MA steel after one year in furnace at 1773 K (courtesy 3-ONE-2, LLC and Kanthal), (**c**) MAX thread bolts used in corrosive environments, (**d**) MAX/Cu vacuum contact materials, and (**e**) MAX/Cu pantographs.

Cold spraying is a relatively new solid-state processing technique in which micron-size solid particles are accelerated to high velocities (500–1000 m/s) towards a substrate with a lower amount of thermal energy (<1000 °C) to produce a coating through complex deformation and bonding mechanisms [12]. To date, not only metal powders, but also ceramic and polymer powders have been successfully deposited on the surfaces of substrates (metals, ceramics and polymers) [12–19]. Cold spraying presents unique advantages, such as the avoidance of powder melting and oxidation, high density, high bond strength, and compressive residual stresses [12].

MAX phase coatings prepared via cold spraying have been extensively studied. The successful cold spray deposition of $Ti_3AlC_2$, $Ti_3SiC_2$, $Cr_2AlC$, and $Ti_2AlC$, as representative MAX coatings, has been reported [20–22]. In contrast to thermal spraying, cold spraying offers attractive properties with respect to the preparation of MAX coatings due to the following typical advantages: (1) MAX particles experience little oxidation and no decomposition upon low-temperature cold spraying, and the resulting coatings retain the original composition and properties of the employed powders; (2) MAX coatings are dense and their bonding strength is high under high-energy impacts; (3) compressive residual stresses rather than tensile stresses are generated in MAX coatings, which is beneficial to a coating's integrity. Cold-sprayed composite coatings containing MAX phases and metals exhibit improved abradable damage tolerance, improved thermal cycling and durability, tailored thermal expansion, and self-lubrication. These coatings are now being used in a number of important applications in turbine engine components, including turbine blades, the vane and the tip of an airfoil, and piston rings [23,24].

This work presents a review of the cold spray deposition of MAX phase coatings. This review principally focuses on the microstructure, mechanical properties, and tribological behaviors of cold-sprayed $Ti_3AlC_2$, $Ti_3SiC_2$, $Cr_2AlC$, and $Ti_2AlC$ MAX coatings. The current limitations and outlooks for future applications in MAX coating research are presented.

## 2. Influencing Factors of Cold-Sprayed MAX Phase Coatings

Some important factors of cold spraying are interrelated and greatly influence the deposition process, quality, and properties of the formed coatings. These major influencing factors include powder characteristics, the driving gas, and the substrate. The influences of such factors on the formation of MAX coatings are discussed as follows.

### 2.1. Powder Characteristics

In the cold-spraying process, the characteristics of the feedstock powder employed, such as particle size, size distribution, and particle morphology (or shape), all influence the resultant coatings.

#### 2.1.1. Particle Size

Feedstock powders are fed into the gas flow and accelerated to high velocity prior to impacting the substrate. A fine powder is easily accelerated to a high velocity and will ease the formation of a denser, thicker coating. However, the price of these powders is a function of their particle size; thus the smaller the powder size, the higher the cost. In addition, if the particles are too small, the bow shock effect due to the presence of a substrate in the spray stream will significantly decelerate their impact speed [12]. Therefore, the particle size of metal powders is usually less than 50 μm for use in cold spraying [12,25]. However, the above particle size requirement is not suitable for ceramic particles because ceramic powders are brittle and thus experience little plastic deformation when impacting a substrate.

Great efforts have been devoted to preparing cold-sprayed MAX phase coatings. The adopted MAX particle sizes are in the range of 0.5–70 μm. Generally, fine MAX particles can achieve a high velocity with which to impact a substrate, thus rendering the coatings thicker and denser. Elsenberg and coworkers [26] prepared different sizes of $Ti_3SiC_2$ MAX powders and discussed the effects of particle size on particle velocity and coating thickness. The initial state of the as-received $Ti_3SiC_2$ powder consists of large particles with an equiaxed shape and smaller ones with a flake-like shape, and has a particle size of D50 = 6.9 μm (Figure 2a). Subsequent high-energy milling led to particle size reduction and morphological change. The D50 particle diameters decreased from 6.2 to 0.5 μm, and their morphology changed from platelet-like to equiaxed shapes after milling for different durations (Figure 2b–e). The authors found that particle size has a profound influence on particle velocity, while other process conditions such as gas flow and pressure have minor influences on particle velocity. High velocities greater than 350 m/s were only achieved for particle sizes smaller than 2 μm. However, the impact velocity of fine particles, especially with sizes smaller than 1 μm, is strongly affected by the bow shock [26,27]. Via optimization of spray conditions, the coating thickness continuously increased with a decreasing mean particle size. At a gas pressure level of 1.1 bar, the best degree of layer buildup, with a thickness of about 500 nm, was observed for the finest powder (D50 = 0.5 μm). Based on their results, it was concluded that a suitable particle size should be optimized to deposit a desirable MAX coating. The use of powder that is too fine may result in a suboptimal coating.

To avoid inducing the bow shock effect on the fine powder and realize the deposition of ceramic coatings, an effective approach is employing agglomerated particles, which are micro-sized but composed of submicron or nanosized grains. For example, Bai et al. [28] prepared spherical $Ti_2AlC$ agglomerated particles with a size of 5–150 μm using spray-drying technology. We have also prepared $Ti_3AlC_2$ and $Cr_2AlC_2$ MAX agglomerated particles. The agglomerated $Ti_3AlC_2$ particles are roughly spherical in shape, and about

75 μm in diameter (Figure 3a). Each agglomerated particle is composed of fine, flake-like $Ti_3AlC_2$ grains. In addition, spherical $Cr_2AlC_2$ particles consisting of small grains with sizes of less than 5 μm have been successfully prepared using spray-drying technology (Figure 3b). These spherical particles have good flowability during cold spraying. If necessary, these agglomerated particles or grains can be further refined via ball milling until they meet the requirements of good cold-sprayed coatings.

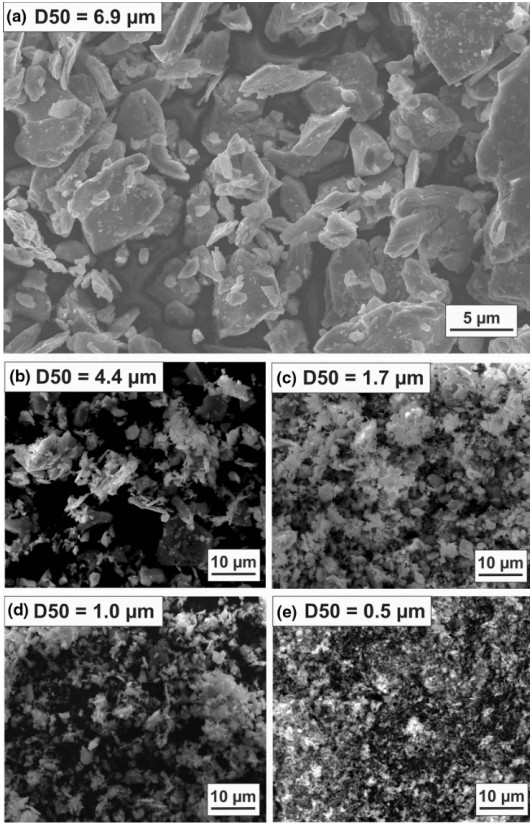

**Figure 2.** SEM images of (**a**) the as-received and (**b**–**e**) ball-milled $Ti_3SiC_2$ powders with different sizes used for cold spraying. Adapted with permission from Ref. [26]. Copyright 2021 Springer Nature.

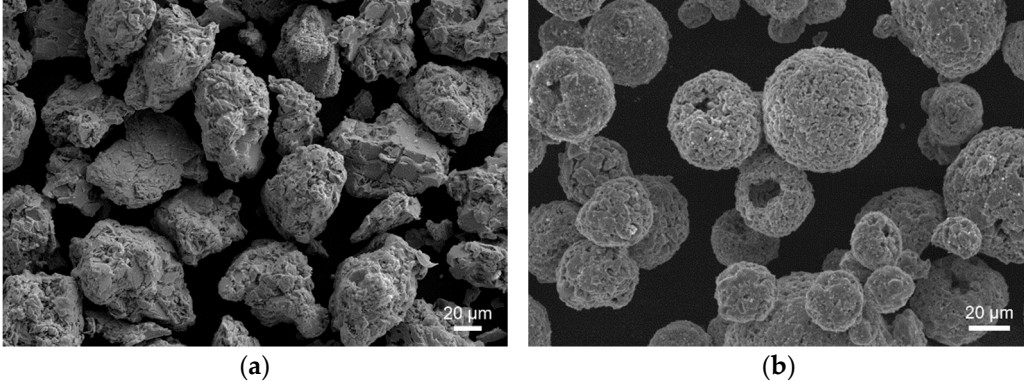

(**a**)          (**b**)

**Figure 3.** SEM images of (**a**) agglomerated $Ti_3AlC_2$ particles and (**b**) agglomerated $Cr_2AlC$ particles.

2.1.2. Particle Size Distribution

MAX phase particles with a single particle size distribution are not suitable for the formation of cold-sprayed coatings; instead, particles comprising a mixture of coarse and fine particles or grains are preferred for the preparation of such coatings. Rech et al. [29] reported that a $Ti_2AlC$ powder with particle sizes of 25~40 μm was used to deposit a coating on an Al substrate via cold spraying. A 50 μm thick $Ti_2AlC$ MAX phase coating

was obtained. Gutzmann et al. [30] prepared a 100 μm thick $Ti_2AlC$ coating via the cold spraying of $Ti_2AlC$ powder with a D50 particle size of 34.3 μm. However, cracks and internal delamination were detected in the coatings due to the limited plastic deformation capacity of $Ti_2AlC$. Many experiments demonstrated that ceramic coatings without cracks and delamination can be achieved via the cold spraying of micron agglomerated-particles containing submicron or nanosized grains. Zang et al. [31] prepared 6 μm-agglomerated $Ti_2AlC$ particles via the hydrothermal treatment of ball-milled $Ti_2AlC$ powder with sizes of less than 0.3 μm. A 100 μm thick and dense $Ti_2AlC$ coating without cracks and delamination was successfully obtained on a Zr-4 substrate by cold spraying the above agglomerated particles at a gas temperature of 400 °C.

### 2.1.3. Particle Morphology

Besides particle size, particle morphology also influences the flowability, the ability to be cold-sprayed, and ultimately, the coating's porosity [12]. In general, spherical particles offer a better flowability than irregular particles. However, previous studies demonstrated that metal particles with an irregular shape can be sprayed at a higher velocity than spherical particles because irregular particles have high drag coefficients [12,32–34]. Thus, irregular particles can be deposited with higher deposition efficiency than spherical particles but may not lead to a lower degree of porosity in the coating since irregular powders are more difficult to pack. The effects of the spherical and irregular morphologies of metal and ceramic powders on a coating's microstructure are complicated. Shockley [35] investigated the influence of $Al_2O_3$ ceramic particle's morphology on coating formation and found that the deposition efficiency (DE) of the irregular $Al_2O_3$ particles was slightly higher than that of the spherical ones. However, the coating containing spherical $Al_2O_3$ particles had improved tribological properties compared to a coating consisting of angular ones. To date, most studies have focused on the cold spray deposition of MAX coatings with irregular particles. The effect of particle morphology on MAX coatings' microstructure and performance should be investigated in the future.

### 2.2. Driving Gases

In the cold-spraying process, the nature, velocity, pressure, and temperature of the driving gases employed have profound influences on the formed coatings. Three types of gases, namely, air, nitrogen ($N_2$), and helium (He), are used as driving gases in cold spraying. The latter two gases are the preferred driving gases due to their inertness. He is the best candidate because it has the lowest molecular weight and can be used to reach the highest velocity among the three gases, but its high cost limits its application in cold spraying. Therefore, $N_2$ is used as the driving gas in cold spraying. Driving gases must be able to be sprayed at high velocities to accelerate feedstock particles with a high enough level of kinetic energy upon impacting a substrate in order to produce a desirable coating. Particles traveling at an insufficient velocity may bounce off the substrate. The impact velocity of particles is generally in the range of 200–1200 m/s. The gas temperature employed can not only increase the gas velocity, but also soften the impact of particles to improve their plastic deformation properties. The driving gas can be heated using an electric heater, for which temperatures range from room temperature to 1200 °C. In addition, high gas pressure provides high drag force for particle acceleration, and affects the gas temperature in the cold-spraying system [36].

During the cold spraying of MAX phase coatings, the pressure and temperature of the $N_2$ driving gas are 3.4–5 MPa and 500–1100 °C, respectively. MAX phase particles can reach a sufficiently high impact velocity with an increasing gas pressure and temperature. Table 1 summarizes the experimental parameters adopted for the cold spray deposition of MAX coatings. Cold-sprayed MAX phase coatings demonstrate the following advantages over thermally-sprayed MAX coatings: no phase transformation, no coating oxidation, and low porosity even at a process temperature of 1100 °C. Elsenberg et al. [21] used $Ti_3SiC_2$, $Ti_2AlC$, and $Cr_2AlC$ MAX phase particles to prepare coatings, and found that $Ti_3SiC_2$ is

unsuitable for cold-spraying due to its brittle characteristics, whereas $Ti_2AlC$ and $Cr_2AlC$ could be cold-sprayed on metal substrates. A $Ti_3SiC_2$ coating is highly non-uniform after cold spraying at a gas pressure of 5 MPa and a gas temperature of 1000 °C (Figure 4a). However, $Ti_2AlC$ and $Cr_2AlC$ MAX coatings can be obtained under the same conditions (Figure 4b,c). $Ti_2AlC$ MAX particles can be easily cold-sprayed into a thicker coating with a size ranging from 300 to 520 μm (Figure 4b). Even at a low temperature of 400 °C, a dense $Ti_2AlC$ coating with a thickness of 100 μm was achieved via the cold spraying of agglomerated $Ti_2AlC$ particles containing submicron grains while using compressed air as the driving gas [31].

**Table 1.** Experimental parameters for cold spray deposition of MAX phase coatings.

| Coating Material | Particle Size (μm) | Substrate Material | Driving Gas | Gas Pressure (MPa) | Gas Temperature (°C) | Particle Speed (m/s) | Stand of Distance (mm) | Thickness (μm) | Remarks | Ref. |
|---|---|---|---|---|---|---|---|---|---|---|
| $Ti_3AlC_2$ | 20–40 | Ti4Al6V | $N_2$ | 3.5–5 | 600–1000 | 723–780 | 30 | 50 | No oxidation at 1000 °C; internal cracks in coatings. | [20] |
| $Ti_3SiC_2$ | 42 (D50) | 304 SS Cu | $N_2$ | 4–5 | 800–1100 | 699–801 | 60 | - | No oxidation; non-uniform $Ti_3SiC_2$ coating. | [21] |
| $Ti_2AlC$ | <62 | Stainless steel | $N_2$ | 3.7 | - | - | 20 | 55 | Continuous transversal cracks in $Ti_2AlC$ coating. | [10] |
| | 11 (D50) | 304 SS Cu | $N_2$ | 5 | 1000 | 717–802 | 60 | 300–530 | No oxidation; low porosity; lateral cracks in coatings. | [21] |
| | <20 | Zr alloy | $N_2$ | 3.5 | 600 | - | - | 90 | No oxidation or phase transformation; low porosity; no delamination in coating. | [22] |
| | 5–150 | TiAl alloy | Air | 1–5 | 100–1000 | - | - | 5–300 | No oxidation; low porosity. | [23] |
| | 5.9 (D50) | Zr alloy | Air | 2–2.5 | 400 | - | 20 | 100 | No oxidation; low porosity. | [31] |
| | 34 (D50) | Cu Stainless steel | $N_2$ | 4 | 600 1000 | - | 60 | 110 155 | No oxidation or phase transformation; low porosity; cracks and internal delamination in coatings. | [30] |
| | 25–40 | Al alloy Stainless steel | $N_2$ | 3.4 3.9 | 500–800 | - | 20 | 50 80 | No oxidation; a continuous transversal crack in coating. | [29] |
| | <20 | Inconel 625 | - | - | - | - | - | 70 | No processing details; voids and microcracks in coating. | [37] |
| | 5–50 | Zr alloy | $N_2$ He | - | 500 800 | - | 26 | 25–30 | Non-uniform coating; microcracks in coating. | [38] |
| $Cr_2AlC$ | 9 (D50) | 304 SS Cu | $N_2$ | 5 | 1000 | 733–849 | 60 | 200–320 | No oxidation; low porosity; lateral cracks in coatings. | [21] |
| | 7.6 (D50) | Stainless Steel | $N_2$ | 4 | 650–950 | - | 60 | 40–97 | No oxidation; low porosity; cracks in coating. | [39] |

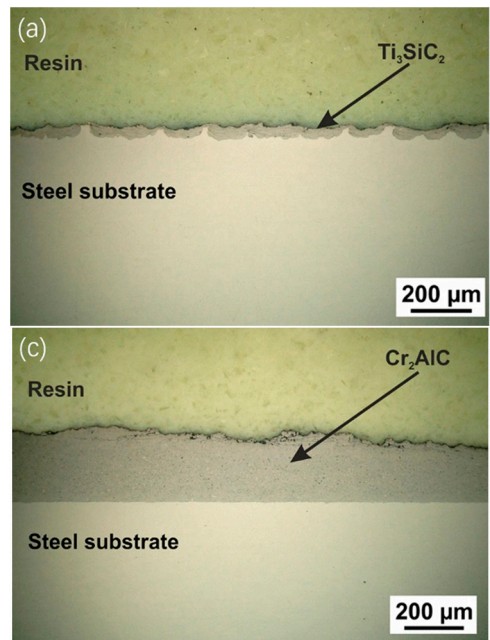
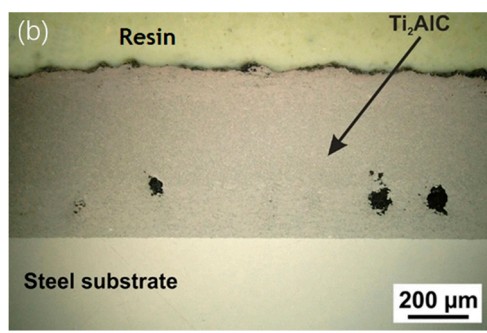

**Figure 4.** Optical micrographs of ten-layer coatings of (**a**) Ti$_3$SiC$_2$, (**b**) Ti$_2$AlC, and (**c**) Cr$_2$AlC after being cold-sprayed on steel substrate with a gas pressure of 5 MPa, a gas temperature of 1000 °C and a traversal speed of 100 mm/s. Adapted with permission from Ref. [21]. Copyright 2020 Springer Nature.

*2.3. Substrates*

So far, cold-sprayed MAX coatings on metal substrates such as Ti, Cu, Zr, Al, Inconel, and stainless steel have been achieved [10,20–22,37–40]. Through plastic deformation, metal substrates mechanically interlock with MAX particles, thus resulting in good adhesion between MAX coatings and metal substrates. However, there is no information on the formation of MAX coatings on ceramic substrates. Cold-sprayed metal particles adhere to the surface of ceramic substrates via the plastic deformation of the particle itself. MAX phases denote ceramic materials with little plastic deformation upon impacting the substrates. It has been reported that cold-sprayed oxide ceramic coatings can be realized on glass or ceramic brittle substrates if the agglomerated particles contain nanostructured crystals [18,41–43]. For example, a dense TiO$_2$ coating with a uniform thickness of several tens of micrometers on an indium tin oxide glass substrate has been achieved via the cold spraying of 25 nm TiO$_2$ particles [43]. Good adhesion between a brittle substrate and nanoparticles may benefit mechanical interlocking. Using these results as an inspiration, MAX coatings on ceramic substrates may be achieved via the cold spraying of nanostructured MAX particles.

**3. Bonding Mechanisms in Cold-Sprayed MAX Phase Coatings**

Understanding the bonding mechanisms involved in cold spraying helps clarify how particles are bonded together and how coatings are built up on the substrate. In this section, microstructural evolution, bonding strength, and residual stresses are discussed to explain the bonding mechanisms in the cold-sprayed MAX coatings.

*3.1. Microstructural Evolution*

During the cold spraying of conventional metal powders, the presence of high temperatures results in the thermal softening of the metal particles, which improves their plastic deformation on the substrate toward the formation of dense coatings. Elsenberg and coworkers [21] investigated the impact morphologies of Ti$_3$SiC$_2$, Ti$_2$AlC, and Cr$_2$AlC MAX powders on 304 steel via cold spraying with a pressure of 4 MPa and a gas temperature of 1000 °C. They found that Ti$_3$SiC$_2$ is the most prone to fracture, in contrast, Ti$_2$AlC

and $Cr_2AlC$ particles show greater plastic deformation at 1000 °C. $Ti_2AlC$ particles' morphologies are shown in Figure 5. At the aforementioned high temperature, most particles adhered to the substrate (marked with "i" and "iii" in Figure 5a). The larger adhering particles are fractured, presenting a nanolamellar structure and cracks (marked with "v" Figure 5b,c). Some smaller particles rebounded and left empty craters (marked with "ii" in Figure 5a,b) in the substrate. Gutzmann and coworkers [30] also reported that most of the impacting $Ti_2A1C$ particles rebounded and left empty craters in the substrate at 600 °C but that increasing the temperature up to 1000 °C improved adhesion. Secondary impacting particles (marked with "iv" in Figure 5b,c) adhered to the primary ones, for which deformation features appeared via slipping along lamellas (marked with "vi" in Figure 5d). A viscous-like deformation effect also occurred along a lamellar failure (marked with "vii" in Figure 5d). Similar impacting morphologies were also observed upon the cold spraying of $Cr_2AlC$ particles [21]. The thermal softening of $Ti_2AlC$ and $Cr_2AlC$ particles at a process temperature of 1000 °C enables good bonding between MAX phase coatings and metal substrates.

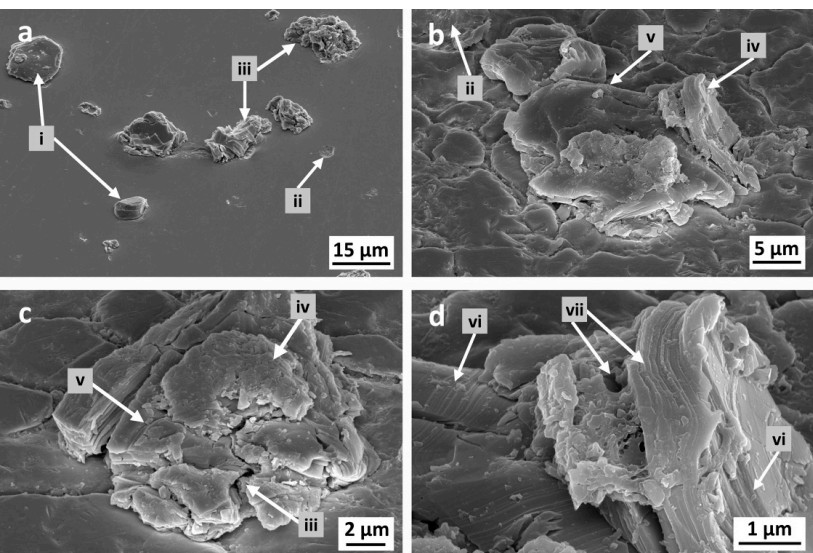

**Figure 5.** Impact morphologies of $Ti_2AlC$ MAX particles on 304 steel substrate after being cold sprayed under conditions of 5 MPa/1000 °C. SEM images with different magnifications (**a**–**d**). The inserts describe the following: i—adhesion of complete, flattened particles, ii—empty impact crater with ASI of steel substrate, iii—fractured particles, iv—secondary particle impacts, v—cracks, vi—deformation on laminae, and vii—viscous-like flow at laminae. Adapted with permission from Ref. [21]. Copyright 2020 Springer Nature.

In cold-sprayed MAX phase coatings, no oxidation or decomposition occurs; thus, the initial composition and structure of the MAX phases are preserved. The thickness of MAX coatings can be tunable, ranging from twenty to five hundred micrometers (see Table 1). The prepared MAX phase coatings are dense and have low porosity. For example, dense, thick $Ti_2AlC$ (Figure 6a) and $Cr_2AlC$ (Figure 6b) on steel were achieved via spraying under conditions of 5 MPa/1000 °C and 4 MPa/950 °C, respectively [21,39]. These coatings were homogeneous, but pores and lateral cracks were detected in the coatings, which deteriorated the mechanical properties of the MAX coatings. However, temperature has a profound influence on porosity and crack concentration; accordingly, the number of pores and cracks decreased with an increasing processing temperature. Porosity decreased from 12.4 vol.% to 9.1 vol.% as temperature increased from 650 to 950 °C [39]. The microstructures shown in Figures 4 and 5 further demonstrate that no cracks or delamination were detected at the interfaces of $Ti_2AlC$/steel and $Cr_2AlC$/steel, suggesting that there was good adhesion between the MAX coatings and the steel substrate.

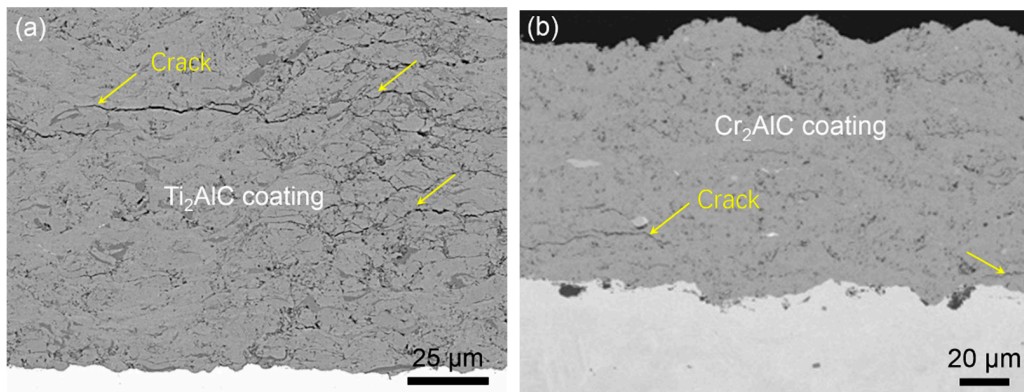

**Figure 6.** Cross-sectional SEM micrographs of (**a**) Ti$_2$AlC coating on steel sprayed under conditions of 5 MPa/1000 °C. Adapted with permission from Ref. [21]. Copyright 2020 Springer Nature. (**b**) Cr$_2$AlC coating on steel sprayed with 4 MPa/950 °C. Cracks are marked with arrows. Adapted with permission from Ref. [39]. Copyright 2018 Elsevier Ltd.

*3.2. Bonding Strength of Cold-Sprayed MAX Phase Coatings*

Adhesion is a fundamental property in a coating/substrate system. The adhesion of cold-sprayed MAX coatings is determined via the bonding of particles to the substrate surface. So far, few studies have focused on the bonding strength between MAX coatings and substrates. Zang et al. [31] reported that a bonding strength of 44 MPa was achieved in a Ti$_2$AlC coating/Zr-4 alloy substrate system prepared via the cold spraying of 0.6 μm Ti$_2$AlC particles. Bonding strength was measured using a tensile adhesion test according to the ASTM C633 standard [44], in which tensile stresses are applied to a coated system consisting of a coated sample glued to another sample. The bonding strength of 44 MPa is even higher than that of 40 MPa of an Al coating/Al alloy substrate [45], 10–20 MPa of Cu coating/steel [46], and 34 MPa of a Ti coating/Al alloy substrate [47] achieved using the same test method.

*3.3. Residual Stresses in Cold-Sprayed MAX Phase Coatings*

Residual stresses are typically compressive stresses occurring on the surface of cold-sprayed coatings due to the peening effect and plastic deformation through the continuous, high-velocity impact of particles. The presence of compressive residual stresses on a coating's surface is beneficial for increasing fatigue strength, resistance to stress corrosion cracking, and bending strength [12]. Go et al. [39] prepared Cr$_2$AlC coatings via cold spraying, and, using X-ray diffraction method, determined that residual stresses were compressive in the coatings. The values of compressive residual stresses increased with increasing process temperature and coating thickness. For example, the compressive residual stresses changed from −200 MPa in the 39.8 μm coating prepared at 650 °C to −310 MPa in the 97.4 μm coatings deposited at 950 °C. However, if the residual stresses are large, they can cause cracking and local delamination in the coatings, or even partial layer spallation [26].

## 4. Mechanical Properties and Tribological Behaviors of Cold-Sprayed MAX Phase Coatings

The reliability of a coating depends on its mechanical properties. The mechanical properties of a cold-sprayed coating are determined by various factors such as microstructure, the adhesion of the coating to the substrate, and residual stress. A dense, fine, and homogeneous microstructure of a cold-sprayed coating is beneficial to the improvement of its mechanical properties. In this section, the hardness and tribological behaviors of MAX coatings will be discussed.

### 4.1. Hardness of Cold-Sprayed MAX Phase Coatings

Hardness is a measure of a material's resistance to localized plastic deformation. It is important to predict the behavior of a coating. The Vickers microhardness test is used to measure a coating's hardness. However, depending on the influence of the substrate, the nano-hardness test is also adopted to measure the hardness of thin coatings.

The micro- and nano-hardness values of different MAX phase coatings, together with those of MAX bulks for comparison, are listed in Table 2. During the cold-spraying process, the high velocity impact of powder induces a refined microstructure, the corresponding cold-working effect, and residual compressive stresses in a coating, leading to an increment in the coating's hardness. The microhardness values of MAX phase coatings are close to or slightly lower than those of MAX bulks. The presence of pores and cracks and a lack of cohesive strength at the particle-particle boundaries negatively impact the hardness of MAX coatings.

**Table 2.** Hardness of MAX bulks and coatings.

| MAX Phase | | GRAIN Size (μm) | Microhardness (GPa) | Nanohardness (GPa) | Ref. |
|---|---|---|---|---|---|
| $Ti_3SiC_2$ | Bulk | 3–200 | 2–6 | 7.3 | [48–50] |
| | Coating | 42 (D50) | 3.75 | 7.9 | [21] |
| $Ti_3AlC_2$ | Bulk | 25 | 3.5 | - | [51] |
| | Coating | 20-40 | - | - | [20] |
| $Ti_2AlC$ | Bulk | 25–50 | 3.3-4.5 | 8.2 | [52–54] |
| | Coating | 11 (D50) | 3.68 | 7.89 | [21] |
| | | 25–40 | - | 10.1 | [29] |
| | | <20 | - | 7–8 | [37] |
| | | <20 | - | 11.8 | [40] |
| $Cr_2AlC$ | Bulk | 2–35 | 3.5–6.4 | - | [55–57] |
| | Coating | 9 (D50) | 5.73 | 11.3 | [21] |

Note: hardness values of MAX coatings are converted into values with a unit of "GPa".

The nanohardness values of MAX phase coatings are greatly higher than the corresponding microhardness values (Table 2). This feature can be attributed to the fact that the indented areas are significantly reduced when the indentation load decreases. When the nanoindentation load is low enough, only small indentations will be formed within a single flattened particle. For example, a microhardness of 3.68 GPa was achieved in the $Ti_2AlC$ coating tested using a Vickers indenter under a load of 1 N, but a nano-hardness value of 7.89 GPa was obtained for the same coating when tested with a Berkovich indenter under a load of 0.01 N [21]. In another study, the nano-hardness value changed from 7 to 15.8 GPa for a $Ti_2AlC$ coating as the load was decreased from 8 N to 0.007 N [37]. This phenomenon is commonly referred to as the indentation size effect (ISE). One of the possible mechanisms of the ISE is that the deformation under the indenter occurs in a discrete manner rather than a continuous one [58].

MAX phases have a hexagonal crystal structure; thus, their nano-hardness value is anisotropic, as the basal planes are parallel or perpendicular with respect to the surface. Concerning a $Ti_3SiC_2$ single crystal with grain sizes of 50–200 μm, Kooi et al. [50] reported that a nano-hardness value of 7.3 GPa was achieved when the basal planes were perpendicular to the surface, which is higher than the value of 4.9 GPa achieved when the basal planes were parallel to the surface, which was due to the greater degree of plastic deformation and the larger pile-up around the indentations for the parallel orientation compared to the perpendicular orientation.

It has been reported that gas temperature, pressure, and particle velocity offer beneficial contributions to the incrementation in the hardness of metal coatings [59,60]. However, there has been little research on the hardness evolution of MAX phase coatings as a function of gas temperature/pressure and particle velocity so far. Therefore, a systematical study

on the relationship between MAX coating hardness and gas temperature/pressure and particle velocity is required.

*4.2. Tribological Behaviors of Cold-Sprayed MAX Phase Coatings*

Wear-resistant coatings on different metallic substrates have been developed to protect metallic components used in different industrial applications. One of the most attractive properties of MAX phases is their self-lubrication [2]. Therefore, a lubricative MAX film or coating can greatly protect substrates from damage by vastly increasing wear resistance and reducing surface friction.

Maier and coworkers [22] prepared a cold-sprayed $Ti_2AlC$ coating on a Zircaloy (zirconium alloy) substrate and investigated its wear resistance. A pin-on-disk wear test was conducted using a 3 mm diameter alumina ball under a 0.02 kg applied load, and a scratch test was performed with a Vickers diamond-tipped scribe under a constant 50 N load. The wear test showed that the wear resistance of the $Ti_2AlC$ coatings was significantly superior to that of the Zircaloy substrate and that the wear track depth decreased from 12 to 1 μm after the cold spray deposition of the $Ti_2AlC$ coating, suggesting enhanced wear resistance. The scratch test demonstrated that scratching did not cause any spallation or delamination of the $Ti_2AlC$ coating from the substrate, indicating good adhesion of the coating to the substrate [22].

Loganathan et al. [37] deposited a $Ti_2AlC$ coating on an Inconel (a nickel-chromium-based superalloy) substrate, which demonstrated enhanced wear resistance both at room temperature and a high temperature of 600 °C. A ball-on-disc wear test was performed using a tribometer, for which an alumina ball of 3 mm in diameter was used as the counter material and a normal load of 5 N and a speed of 100 RPM were applied in dry air both at room temperature (25 °C) and 600 °C. The wear test showed that the average coefficient of friction (COF) of the coating was 0.767 at room temperature but 0.603 at 600 °C. The wear depth decreased from 25 m at room temperature to 12 μm at 600 °C, and the calculated wear rates also changed from $4.7 \times 10^{-7}$ $mm^3$/Nmm at room temperature to $2.84 \times 10^{-7}$ $mm^3$/Nmm at 600 °C, constituting a 40% decrease. The lower COF and wear rate at 600 °C resulted from the fact that a mostly uniform and continuous oxide tribofilm consisting of $TiO_2$ and $Al_2O_3$, which are relatively lubricious at high temperature, formed at 600 °C and provided a lubricious effect. The wear mechanism of the cold-sprayed $Ti_2AlC$ coating changed from brittle at room temperature to ductile at a high temperature. The above results suggested that cold-sprayed $Ti_2AlC$ coatings on metal substrates have great potential applications at high temperatures up to 600 °C.

## 5. Summary and Outlook

This review has presented the recent progress on the cold spraying of MAX phase coatings. The influencing factors on the formation, bonding mechanisms, mechanical properties, and wear resistance of MAX phase coatings have been discussed. It is clear that cold spraying is a valid technique for developing MAX phase coatings. This work provides some basic information for the future preparation and research of cold-sprayed MAX phase coatings and accelerates the practical applications of such coatings. However, there are still some issues that have not been comprehensively explained, and future work is still needed to obtain such knowledge. Future directions and research gaps are listed as follows:

(1)     Powder size and morphology

Irregular MAX phase particles with sizes of 0.5–70 μm have been used to prepare MAX coatings via cold spraying. The influences of particle size (especially in a nano scale) and morphology (irregular and spherical shapes) on the microstructure and properties of cold-sprayed MAX coatings have received less attention and should thus be further investigated.

(2)     Interface characterization

MAX phase coatings can be cold-sprayed on metal substrates, for which good adhesion is achieved. However, the absence of interfacial characterization renders the nature of the bonding mechanisms between MAX coatings and metal substrates ambiguous. Interfacial features influence bond strength. One of the most readily identifiable interfacial features is interfacial melting. Interfacial melting occurs during the cold spraying of some metal particles on metal substrates, wherein the high energy impact of cold spray particles can render the particles or the substrate molten. Large splashes and droplets can be seen to radiate from bonded particles and rebound sites, and nanosized, spheroidized droplets can also be detected at particle-particle interfaces, contributing to high bond strength [61–63]. As MAX particles impact metal substrates with high velocity, does interfacial melting occur between the coating and the metal substrate or at particle-particle interfaces? In addition, are there interfacial amorphous layers between MAX coatings and metal substrates due to strain-rate-induced amorphization? At the bonded interface, microstructural inhomogeneity must also be characterized via electron microscopy.

(3)    Bonding mechanisms

Cold spray bonding is a combination of mechanical interlocking and metallurgical bonding. Recrystallization, atomic diffusion, and interfacial melting contribute to the metallurgical bonding mechanisms for cold-sprayed metal coatings. At present, mechanical interlocking is considered to be the predominant bonding mechanism between MAX phase coatings and metal substrates. The metallurgical bonding mechanism for cold-sprayed MAX phase coatings has been discussed to a lesser extent. The detailed bonding mechanisms that affect particle-to-substrate adhesion and particle-to-particle adhesion are still unclear. Research on bonding, especially in the MAX phase particles, is required.

(4)    Expand the types of MAX phase coatings and substrates.

So far, only four kinds of cold-sprayed MAX phases ($Ti_3AlC_2$, $Ti_3SiC_2$, $Cr_2AlC$, and $Ti_2AlC$) coatings have been deposited on metal substrates. In the MAX phase family, about one hundred types of compounds with different functional properties have been synthesized. It is necessary to develop different cold-sprayed MAX phase coatings with tailored properties for specific applications. In addition, several metal substrates with cold-sprayed MAX phase coatings have been investigated. In the future, the cold spraying of MAX phase coatings on metal, ceramic, and polymer substrates is necessary in order to meet practical application requirements.

(5)    Computational simulation of cold-sprayed MAX phase coatings.

Computational simulation plays an important role in the development of coatings and the understanding of cold spray bonding. Great efforts have been devoted to the computational simulation of cold-sprayed coatings on different substrates. Computational simulation is also required to analyze the influencing factors with respect to the quality of MAX phase coatings and to understand their bonding mechanisms.

(6)    Performance of cold-sprayed MAX phase coatings

The mechanical and wear resistance properties of cold-sprayed MAX phase coatings have been investigated. Future work on high-temperature oxidation resistance, corrosion resistance, electrical properties, and other properties of MAX phase protective coatings, and on the relationship between influencing factors and the performance of cold-sprayed MAX phase coatings, is required.

(7)    The post-heat-treatment is essential for coating consolidation and microstructure modification.

The influence of heat treatment on the microstructure and properties of MAX phase coatings has not been assessed, and this is required for further investigation.

**Author Contributions:** Investigation, W.Z., X.Z. and X.C.; Supervision, S.L.; Writing—Original Draft Preparation, W.Z.; Writing—Review and Editing, S.L.; All authors have read and agreed to the published version of the manuscript.

**Funding:** This work was supported by the National Natural Science Foundation of China under Grant No. 52275171, and the Pre-Research Program of the National 14th Five-Year Plan (No. 80923010304).

**Institutional Review Board Statement:** Not applicable.

**Informed Consent Statement:** Not applicable.

**Data Availability Statement:** Not applicable.

**Conflicts of Interest:** The authors declare no conflict of interest.

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
