# Peer review of "Research and Development on Cold-Sprayed MAX Phase Coatings"

_coatings, doi:10.3390/coatings13050869_

Round 1

Reviewer 1 Report

Comments to authors :

Reviewer: General Comments: The review describes Research and development on cold-sprayed MAX phase coatings.

I think this review is acceptable after some revisions.

The listed modifications are proposed before reconsideration:

1.     Some patents should also be cited and reviewed.

2.     More works can be incorporated in the Table since it is a review article.

3.     Please highlight the most significant findings in conclusions. Please also provide future perspectives and research gap.

 Recommandation : Publish after minor revisions. Summing up, I do advise publication of this work in this form, the paper requires minor revisions before it can be published in Coatings.

Author Response

Reviewer: 1

Comments: Publish after minor revisions. Summing up, I do advise publication of this work in this form, the paper requires minor revisions before it can be published in Coatings. 

Thanks for the reviewer’s constructive comments.

1) Some patents should also be cited and reviewed.

Thank you for your good suggestion. Some patents on the cold-sprayed MAX phase coatings were also cited and relative contents were added in the text.

2) More works can be incorporated in the Table since it is a review article.

Thank you for your good suggestion. So far, only Ti3AlC2, Ti3SiC2, Cr2AlC and Ti2AlC MAX ceramic coatings have been studied. Therefore, almost data on the MAX coatings were summarized in Table 1. In the future, a large number of MAX phases can be used to prepare cold-sprayed coatings.

3) Please highlight the most significant findings in conclusions. Please also provide future perspectives and research gap.

Thank you for your good suggestion. The significant findings, future perspectives and research gap were summarized in conclusions.  

Reviewer 2 Report

The review article summarizes recent advancements in cold-sprayed of the MAX phase coatings. The review is well written and informative. Several factors affecting the properties of the coatings are discussed and well-illustrated with examples taken from the literature. However, for the purpose of the review only a limited number of coatings is discussed, namely Ti3AlC2, Ti3SiC2, Cr2AlC and Ti2AlC MAX coatings. I would advise to consider narrowing of the article title, so that it better corresponds to the type of coating materials revised. Further, I would recommend making a more thorough literature and patent research regarding the subject – majority of the cited references are older than from the last 5 years. Were there any new discoveries or advancements done in recent years?

Editorial remark: lines 162-163: The reference to the cited studies is missing in the phrase: “Shockley investigated 162 the influence of Al2O3 ceramic particle morphology on the coating formation...”. Please add.

Author Response

Reviewer: 2

The review article summarizes recent advancements in cold-sprayed of the MAX phase coatings. The review is well written and informative. Several factors affecting the properties of the coatings are discussed and well-illustrated with examples taken from the literature.

Thanks for the reviewer’s constructive comments.

However, for the purpose of the review only a limited number of coatings is discussed, namely Ti3AlC2, Ti3SiC2, Cr2AlC and Ti2AlC MAX coatings. I would advise to consider narrowing of the article title, so that it better corresponds to the type of coating materials revised.

Thanks for the good suggestion. So far, only four kinds of MAX ceramic coatings, i.e. Ti3AlC2, Ti3SiC2, Cr2AlC and Ti2AlC, have been investigated. These phases all belong to the MAX phase family. The MAX phase coatings demonstrate attractive properties. In the future, many types of MAX phase coatings will be prepared and their properties will be demonstrated.

This review presents the recent progress on the cold -sprayed MAX phase coatings, and aims to attract more researchers' attention to the MAX coating and perform related research.       Further, I would recommend making a more thorough literature and patent research regarding the subject – majority of the cited references are older than from the last 5 years. Were there any new discoveries or advancements done in recent years?

Thanks for the good suggestion. Patents on the cold-sprayed MAX phase coatings were also added in the text. Recent discoveries were all listed in this work.

Round 2

Reviewer 2 Report

Authors sufficiently responded to my comments.